

# Technology-assisted white cane: evaluation and future directions

Izaz Khan, Shah Khusro and Irfan Ullah

Department of Computer Science, University of Peshawar, Peshawar, Khyber Pakhtunkhwa, Pakistan

## ABSTRACT

**Background**. Several technology-assisted aids are available to help blind and visually impaired people perform their daily activities. The current research uses the state-of-the-art technology to enhance the utility of traditional navigational aids to produce solutions that are more reliable. In this regard, a white cane is no exception, which is supplemented with the existing technologies to design Electronic Travel Aids (ETAs), Electronic Orientation Aids (EOAs), and Position Locator Devices (PLDs). Although several review articles uncover the strengths and limitations of research contributions that extend traditional navigational aids, we find no review article that covers research contributions on a technology-assisted white cane. The authors attempt to fill this literature gap by reviewing the most relevant research articles published during 2010–2017 with the common objective of enhancing the utility of white cane with the existing technology.

**Methods**. The authors have collected the relevant literature published during 2010–17 by searching and browsing all the major digital libraries and publishers' websites. The inclusion/exclusion criteria were applied to select the research articles that are relevant to the topic of this review article, and all other irrelevant papers were excluded. Among the 577 (534 through database searching and 43 through other sources) initially screened papers, the authors collected 228 full-text articles, which after applying exclusion/inclusion criteria resulted in 36 papers that were included in the evaluation, comparison, and discussion. This also includes research articles of commercially available aids published before the specified range.

**Results**. The findings show that the research trend is shifting towards developing a technology-assisted white cane solution that is applicable in both indoor and outdoor environments to aid blind users in navigation. In this regard, exploiting smartphones to develop low-cost and user-friendly navigation solution is among the best research opportunities to explore. In addition, the authors contribute a theoretical evaluation framework to compare and evaluate the state-of-the-art solutions, identify research trends and future directions.

**Discussion**. Researchers have been in the quest to find out ways of enhancing the utility of white cane using existing technology. However, for a more reliable enhancement, the design should have user-centric characteristics. It should be portable, reliable, trustworthy, lightweight, less costly, less power hungry, and require minimal training with special emphasis on its ergonomics and social acceptance. Smartphones, which are the ubiquitous and general-purpose portable devices, should be considered to exploit its capabilities in making technology-assisted white cane smarter and reliable.

Corresponding author
Shah Khusro, khusro@uop.edu.pk

## INTRODUCTION

The white cane is one of the conventional navigation aids and a symbol of identification for blind people. It is low cost, reliable, efficient, simple, and allows direct physical interaction with the ground through signaling effects. It extends the sensing capabilities of blind people to better understand the surrounding environment by enabling them to adopt various techniques including obstacle detection, echolocation, and shorelining. It enables the blind user to detect obstacles in the range of 1.2 m at ground level. A blind user sweeps the white cane to the left and right while walking (in the forward direction) so that they can detect steps, drop-offs, and curbs and find if running into the obstacles. Sweeping the white cane (dragging and tapping) techniques help them understanding the objects' nature including shape, hardness, dimensions, etc., in creating their mental maps of the surroundings (*Rosen, 2018*). Echolocation enables blind users in navigation using echoes to identify objects with sound-reflecting surfaces such as walls, parked cars, etc. (*Brazier, 2008*). Echolocation gives the perception about the surrounding environment in the form of the object, its location, dimensions, and density and exploits the variation in the sound of the individual (*Brazier, 2008*). Shorelining is a trailing or tracking technique used by the blind people to follow the edge or "shoreline" of the travel path. This technique is often suitable when the user wants to find a specific item or location along the edge of the travel path such as finding intersecting hallways, mailbox or sidewalk. Users also employ this technique to maintain their contact with specific landmarks to go from one point to another and for avoiding open spaces. For shorelining the blind people usually follow a border of (wall, corridor) or pavement marks specially created for the blind user in public places such as at bus stops, shopping center, crosswalk (*Rosen, 2018*) .

Besides these essential techniques used by blind people to benefit from the white cane, they are unable to detect trunk and head-level obstacles and other over-hanging objects (*Pyun et al., 2013*). It requires direct physical contact with the obstacle and unable to inform the user about the approaching obstacles (*Wang & Kuchenbecker, 2012*). With the inception and proliferation of ubiquitous sensing and computational technology, researchers have been able to overcome these limitations by supplementing the basic functionalities of white cane to enable blind people to perform most of these navigation-related activities independently. Researchers have exploited technology in developing technology-based navigational and orientation aids, which based on their application domain, can be categorized into Electronic Travel Aids (ETA), Electronic Orientation Aids (EOAs), and Position Locator Devices (PLDs) (*Nieto, Padilla & Barrios, 2014*). ETAs use sensors to collect sensory data and use it in navigation (*Leduc-Mills et al., 2013*; *Gallo et al., 2010*; *Vera, Zenteno & Salas, 2014*).   EOAs provide orientation aid to disabled users in navigation. PLDs use Global Positioning System (GPS) and other techniques in localization and tracking of blind users. Most of these aids have exploited white cane as the primary

navigation tool and supplemented it with the state-of-the-art technologies for improved immediate tactile information about the ground, drop-offs, direct physical interaction, and signaling effect with surroundings.

Several state-of-the-art review articles (*Fallah et al., 2013*; *Tapu, Mocanu & Tapu, 2014*) cover research and application technologies related to white cane. *Fallah et al. (2013)* present a comprehensive review of indoor navigation systems and highlight the techniques used in path planning, user localization, environment presentation and user-system interaction. *Tapu, Mocanu & Tapu (2014)* survey the wearable systems used by blind users in outdoor environments and identify the advantages and limitations of each of the studied system. Several performance parameters were introduced to classify the reported systems by giving qualitative and quantitative measures of evaluation. *Kim et al. (2016)* conducted a user study on the characteristics of long cane that the researchers may consider potentially in developing ETAs. They focused on the orientation and sweeping frequency of long cane while it is in constant contact with the ground. They measured the average sweeping frequency, tilt angle, and grip rotation deviation of a long cane. Besides minor variation among users, for about 90% of the subjects, the index finger and thumb are in contact with cane handle. Also, they found significant differences among the subjects regarding sweeping range with low variations in frequency.

Besides significance and valuable research implications, the scope of these survey and review articles is limited only to the technologies used for navigation in indoor and outdoor environments and the pros and cons of these devices such as power consumption, intrusiveness, and user acceptability. These studies are generic covering some general aspects including, e.g., localization of the user in the environment, path-planning techniques, user-system interaction, device operation, sensor information translation, and unable to cover research articles that have extended white cane with technology to develop low-cost and user-friendly navigation solutions. The only exception is *Kim et al. (2016)*. However, it is a user study on long cane sweeping angle, tilt angle, and grip rotation. Also, because of its scope, it is unable to cover other factors that are mandatory for the design of suitable ETAs.

This review article fills this literature gap to understand the potential role of white cane in developing user-friendly navigation aids. The paper evaluates some of the significant aspects of navigation aids including operating environment, device sensing range, computational device, information presentation techniques, and so on, which can be exploited in developing more efficient and effective technology-assisted white cane. The paper also identifies the current trends from the relevant literature published during 2010–17 and contributes some future research directions. More specifically, the contributions of the paper include:

- To identify the state-of-the-art in technology-assisted white cane for obstacle detection, navigation, and orientation by reviewing relevant research publications published during 2010–2017.
- To contribute a theoretical evaluation framework for comparing the existing technology-assisted white cane solutions, identify current research trends, and possible future directions.

'Survey Methodology' of this paper presents survey methodology. 'Summary of Key Observations' presents the evaluation framework, which is used to evaluate the selected 36 papers and summarizes findings based on this evaluation. It also uncovers the research trends and future directions. 'Conclusions' concludes our discussion. References are presented at the end.

## SURVEY METHODOLOGY

### Literature search, inclusion/exclusion criteria

For collecting the relevant literature, published in the last eight years (2010–17), the authors followed *Moher et al. (2009)*. The authors searched and browsed Google Scholar, Springer, ScienceDirect, ACM digital library, and IEEE. We tried different search queries including "white cane", "virtual white cane", "technology-assisted white cane", "assistive tools for visually impaired and handicapped", "blind assistive technologies", and "navigation aids", and kept the publication year as 2010–17. In the search results against each query, each entry was scrutinized to check whether the query terms appear in the title or snippet and, if yes, was opened in new tab to read in detail its title, abstract, and references (if available). If deemed relevant, the full-text of the article was downloaded to apply our inclusion/exclusion and eligibility criteria. Also, the authors performed the manual inspection of the references list of each selected full-text publication to identify relevant and related papers. This includes research papers that were published beyond the specified range but are still mass produced or commercially available. We adopted the PRISMA checklist, adapted from *Moher et al. (2009)* and depicted in Fig. 1, at different stages of our exploration, from search to the final selection of full-text articles to include in the literature review. Our *eligibility and inclusion* criteria consider research publications that:

- enhance the utility of white cane with the existing technologies either for indoor, outdoor or both types of operating environments
- are published beyond the specified range but their developed ETAs are still mass produced or commercially available
- place the sensor unit on the white cane
- discuss virtual white cane, devices that mimic the attributes of the traditional white cane to assist blind users.

The *exclusion criteria* include articles that:

- are unable to assist blind users through technology
- discuss assistive devices, which are unable to mimic or extend white cane for navigation
- have less academic significance or with unauthorized publishers or repetitive ideas and thus lack in novelty or originality.

The inclusion/exclusion criteria resulted in 36 articles. Also, there are two systems whose publications are unavailable, and therefore, their details are included from their official websites. This results, in total, 38 systems, which are included for comparison and evaluation using the proposed evaluation framework and in the identification of research trends and future research opportunities.

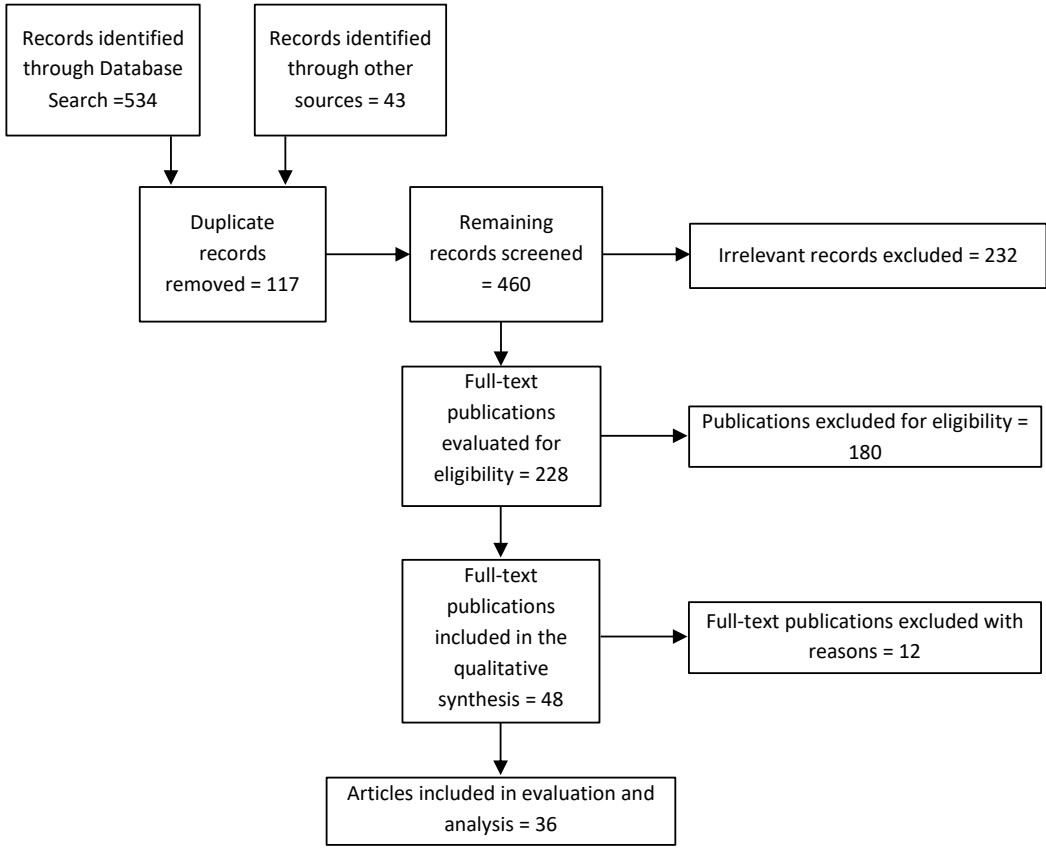

**Figure 1**   **PRISMA Flow diagram (adapted from *Moher et al., 2009*).**

## Evaluation criteria and framework

Figure 2 presents the nine criteria that we have identified from the selected 38 ETAs. These include (i) operating environments; (ii) use of sensors (single, multiple); (iii) positioning of sensory unit on white cane; (iv) sensor covering range and area; (v) mode of operation; (vi) computational devices; (vii) functionality; (viii) localization techniques; and (ix) user-system interaction (input/output). Table 1 summarizes each of these criteria with their purpose, possible values, and representative symbols, which are then used in Table 2 ('Summary of Key Observations') to evaluate and compare the state-of-the-art technology-assisted white cane solutions along with the results and research implications.

## SUMMARY OF KEY OBSERVATIONS

This Section uses the evaluation framework (Table 2) to identify trends and possible future directions. Table 2 evaluates and compares the state-of-the-art solutions that meet the eligibility and inclusion criteria discussed in 'Survey Methodology'. It also includes the assistive tools that have no corresponding paper but are discussed and detailed on their respective official websites. In this case, all the required details were collected from their websites. The identified research trends are further visualized in Fig. 3.

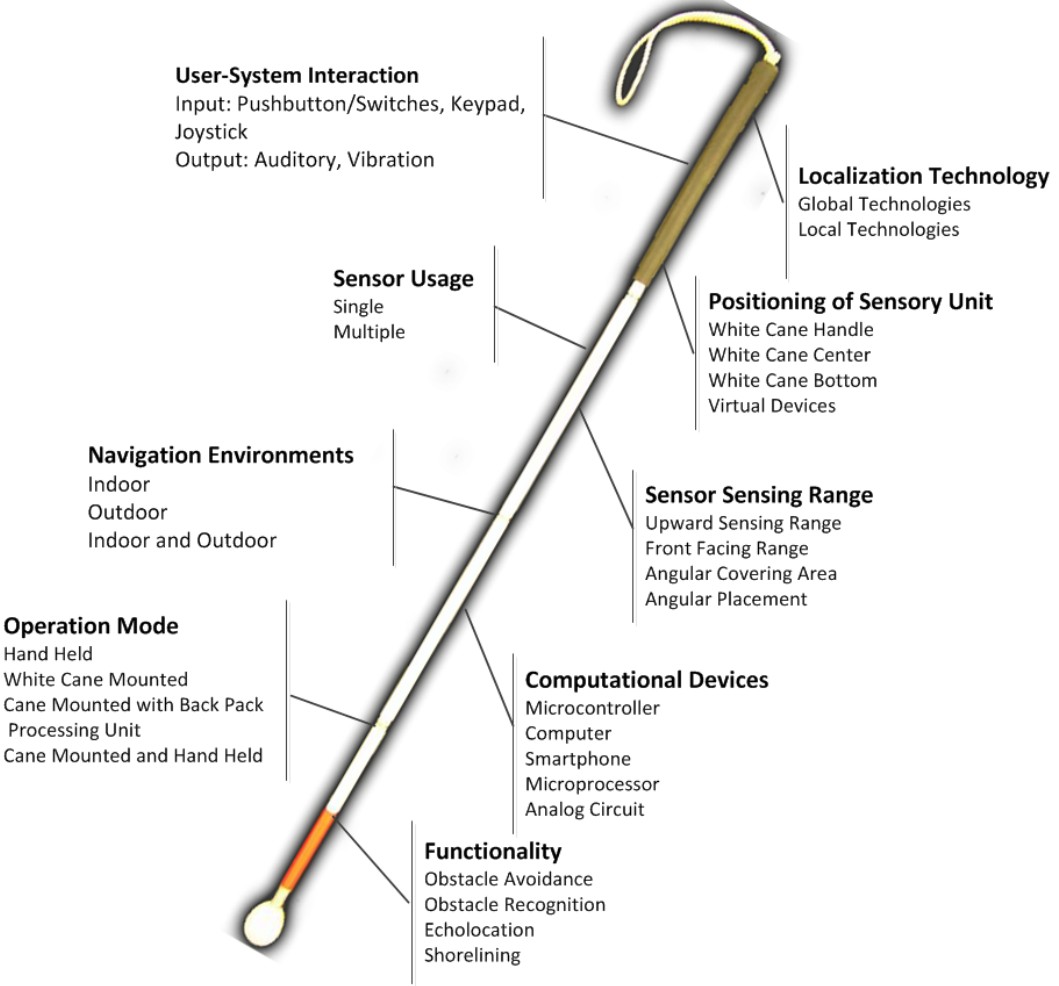

**Figure 2   The identified factors in technology-assisted white cane.**

In addition to the key observations discussed in the following nine criteria, other aspects include cost, power consumption, weight, size, shape, ergonomics, and social acceptance. These aspects are briefly discussed from the perspective of each of these nine criteria in 'Other factors'. The reason behind keeping the discussion in a separate subsection under is the inter-relatedness and interdependence of these factors on each other as well as on the nine factors discussed in this Section. In most of the studies in Table 2, these aspects are not mentioned explicitly, and therefore, the authors dug out these implicit details for inclusion in the article.

## Operating environments

Technology-assisted white cane has been designed from its usage perspective either for the indoor environment—enclosed areas including small and large-buildings (*Koyuncu & Yang, 2010*), outdoor environment—spaces outside these buildings including streets, playgrounds, roads, etc., (*Hersh & Johnson, 2008*) or both (Table 2, column 3). Figure 3A

**Table 1  The Criteria for evaluating technology-assisted white cane.** The authors use abbreviations/mnemonics to represent these entries in Table 2 for clarity and adjusting the specified details into the corresponding cells.

| Evaluation Criteria | Purpose | Values/Symbols |
|---|---|---|
| Operating Environment (OE) | To differentiates the state-of-the-art technology-assisted white canes based on the environments for which they are designed to operate in | Indoor (I), Outdoor (O), Indoor and Outdoor (IO) |
| Sensors | To identify the number, type, facing of sensors exploited in white canes for sensing purposes | Single Sensor (S), Multiple Sensor (M) |
| | | Sensor Type (T): Ultrasonic (US), Laser (L), Gyro (G), Pedometer (P). Infrared (IR), Camera (Cm), Accelerometer (Acc), Water (W), Magnetometer (M) |
| | | Sensor Facing (SF): Upward (U), Downward (D), Left L, Right R, Front F. arbitrary (O), Trunk Level (T), Drop Offs (DO) |
| System Functionality | To describe the desired functionality of the technologically assisted white cane regarding obstacle detection, obstacle recognition, echolocation and shorelining; | Obstacle Detection (OD), Obstacle Recognition (OR), Shorelining (S), Echolocation (E). |
| Positioning of Sensory Unit (PSU) | To describe the placement of sensors in the different areas/parts of a white cane | On White Cane Handle (WCH), On White Cane Center (WCC), On White Cane Bottom (WCB), Virtual White Cane (VWC) |
| Sensor Sensing Range | To describe the sensing range of sensors in different directions | Front Range (FR), Upward Range (UR), Angular Covering Area/ Angular Placement(ACA/AP) |
| Operation Mode (OM) | To classify the technology-assisted white cane based on their operation mode | Handheld (HH), Cane Mounted with Back Pack, Computation Unit (CMBPCU), Cane Mounted (CM), Head Mounted (HM), Cane Mounted and Handheld (CMHH) |
| Computational Device (CD) | To describe the type of data processing and decision-making device | Microcontroller (MC), Smartphone (SP), Analog Circuit (AC), Microprocessor (MP), Computer (C), Google Tango (GT), Not Given (NG) |
| Localization Technology (LT) | To describe the technology used for localization in the surroundings | Global Positioning System (GPS), Radio Frequency Identification (RFID), Quick Response Codes (QR-Codes) |
| User-System Interaction (USI) | To identify how users interact with a proposed solution in the form of input and output from the system | Input (I): Push Buttons/Switches (PB/S), Keypad (KP), Joystick (JS) Output (O): Auditory (A), Vibration (V), Auditory and Vibration (AV) |

depicts that most of the existing systems are for both indoor and outdoor environments. It means that the trend is to design and develop a generalized technology-assisted white cane, which can adapt itself to the changing environment and keep the users away from changing/choosing apps that treat each environment differently. This adaptation is required to reduce the cost, training, and cognitive load on the blind users as they would not have to buy, use and become used to with several devices.

## The use of sensors

Several different types of sensors have been used to assist blind people in navigation. Their selection depends upon the nature of the problem and the corresponding operating environment. In the selected studies, either a single sensor was used or multiple sensors were fused together (see Table 2, column 4). *Wang & Kuchenbecker (2012)* use only ultrasonic

Khan et al. (2018), *PeerJ*, DOI 10.7717/peerj.6058

**Table 2  Evaluation and comparison of technology-assisted white cane solutions.**

| S. No. | Study/Name of the system | OE | Sensors | | | PSU | Sensor Sensing Range | | | OM | CD | System Functionality | | | | LT | User-System Interaction | |
|---|---|---|---|---|---|---|---|---|---|---|---|---|---|---|---|---|---|---|
| | | | S/M | T | SF | | FR | UR | ACA/AP | | | OD | R | E | S | | I | O |
| 1. | ARIANNA (*Croce et al., 2014*) | I | S | Cm | F | VWC | × | × | × | HH | SP | × | × | E− | S− | QR | × | V |
| 2. | Electronic Cane (*Bouhamed et al., 2012*) | I | M | US, Cm | U-D, F | WCC | 0.03–10 m | × | × | CM | MC | ✓ | ✓ | E− | S− | × | × | A |
| 3. | Kinect Cane (*Takizawa et al., 2012*) | I | M | Cm, IR, Acc | F | WCH | 0.1–0.5 m | × | 35°/× | CMBPCU | C | ✓ | ✓ | E− | S | × | KP | AV |
| 4. | (*Niitsu et al., 2012*) | I | M | US, G, BT, Comp | F | WCH | 4 m | 4 m | × | CM | SP | ✓ | × | E− | S− | × | × | A |
| 5. | Co-Robotic Cane (*Ye, Hong & Qian, 2014*), | I | M | Cm, G | F | WCH | 5 m | × | × | CM | C | ✓ | ✓ | E− | S+ | × | KP | RG, A |
| 6. | CCNY Smart Cane (*Chen et al., 0000*) | I | M | IR, Cm, G, Acc | F | WCH | 0.5 m | × | × | CM, | MC, GT | ✓ | ✓ | E− | S+ | × | JS, PB/S | AV |
| 7. | (*Munteanu & Ionel, 2016*) | I | M | US, BT | F, U | VWC | 0.2–4 m | 0.2–4 m | × | HH | MC | ✓ | × | E− | S− | × | × | AV |
| 8. | (*Ye, 2010*) | I | S | Cm (3D) | F | WCH | 0–5 m | × | 34–43°/× | CM | C | ✓ | ✓ | E− | S− | × | × | AV |
| 9. | (*Alshbatat & Ilah, 2013*) | O | M | US, GPS | F, O | WCH | 5 m | × | × | CM | MC | ✓ | × | E− | S− | GPS | KP | AV |
| 10. | (*Madulika et al., 2013*) | O | M | US, GPS | F, O | WCH | 0.02 m–6.5 m | × | × | CM | MC | ✓ | × | E− | S− | GPS | × | AV |
| 11. | *Andò et al. (2015)* | O | M | US, Acc | L-R, O | WCC | 0.8 m | × | × | CM | MC | ✓ | × | E− | S− | × | × | V |
| 12. | (*Mutiara, Hapsari & Rijalul, 2016*) | O | M | US, Acc, G, M | F, D | WCH, WCB | 0.03–1.5 m | × | ×/45° | CM | MC | ✓ | × | E− | S+ | × | PB/S | A |
| 13. | Ultra-Body Guard (*RTB, 2018*) | O | M | US, Light, Comp | F, U | WCH | 3 m | 3 m | × | CM | MP | ✓ | ✓ | E− | S− | × | × | AV |
| 14. | SmartCane (*Vaibhav et al., 2010*) [a] | IO | S | US | F, U | WCH | 3 m | 3 m | 1.2–1.3 m/50°–60° | CM | MC | ✓ | × | E− | S− | × | × | V |
| 15. | Miniguide [b] (*Hill & Black, 2003*) | IO | S | US | F, U | VWC | 3 m | 3 m | × | HH | MC | ✓ | × | E− | S− | × | × | V |

| S. No. | Study/Name of the system | OE | Sensors | | | PSU | Sensor Sensing Range | | | OM | CD | System Functionality | | | | LT | User-System Interaction | |
|---|---|---|---|---|---|---|---|---|---|---|---|---|---|---|---|---|---|---|
| | | | S/M | T | SF | | FR | UR | ACA/AP | | | OD | R | E | S | | I | O |
| 16. | UltraCane [c] (Withington et al., 2004) | IO | S | US | F, U | WCH | 4.5 m | 4.5 m | × | CM | MC | ✓ | × | E− | S− | × | × | V |
| 17. | Palm Sonar [d] (Corporation, 2018) | IO | S | US | F, U | VWC | 0.7–4 m | 0.7–4 m | 30°/0 | HH | MC | ✓ | × | E− | S− | × | PB/S | V |
| 18. | Ray [6] (Smith & Penrod, 2010) | IO | M | US, Light | F, U | VWC | 1.7–2.5 m | 1.7–2.5 m | 30°/0 | HH | MC | ✓ | × | E− | S− | × | PB/S | AV |
| 19. | (Nieto, Padilla & Barrios, 2014) | IO | M | US, GPS | F, O | VWC | 0.2–2.5 m | 0.6–2 m | × | HH | MC, SP | ✓ | × | E− | S− | GPS | PB/S | AV |
| 20. | (Vera, Zenteno & Salas, 2014) | IO | M | L, Cm | F | VWC | 0.3–1.7 m | × | × | HH | SP | ✓ | × | E− | S− | × | × | V |
| 21. | (Pyun et al., 2013) | IO | M | US, IR | F-U-T, DO | WCH | 3.5 m | 3.5 m | × | CM | MC | ✓ | × | E− | S+ | × | PB/S | V |
| 22. | HALO (Wang & Kuchenbecker, 2012) | IO | S | US | F | WCH | 1.83 m | × | × | CM | AC | ✓ | × | E+ | S− | × | × | V |
| 23. | (Mahmud et al., 2014) | IO | S | US | L−R−F | WCB | 0.1–2 m | × | × | CM | MC | ✓ | × | E− | S− | × | × | AV |
| 24. | (Nada et al., 2015) | IO | M | US, IR, W | F-U, D | WCH, WCB | 4 m | 4 m | × | CM | MC | ✓ | × | E | S | × | × | AV |
| 25. | Smart Cane (Wahab et al., 2011) | IO | M | US, W | F, D | WCH | 1 m | × | × | CM | MC | ✓ | × | E− | S− | × | × | AV |
| 26. | (Okayasu, 2010) | IO | M | US, Cm | F-U, F | WCH | 0.5–5.5 m | 0.5–5.5 m | × | CM | NG | ✓ | × | E− | S− | × | × | V |
| 27. | (Gallo et al., 2010) | IO | M | US, IR | F-U, F | WCH | 5.5 m | 0.4–1 m | × | CM | MC | ✓ | × | E+ | S− | × | × | AV |
| 28. | (Kassim et al., 2013) | IO | S | RFID | O | WCB | 0.037 m | × | × | CM | MC | ✓ | × | E− | S− | RFID | × | A |
| 29. | NavEye (AlAbri et al., 2014) | IO | M | US, Cm, BT | F | WCB | <1 m | × | × | CM,HH | MC, SP | ✓ | × | E− | S− | QR | PB/S | AV |
| 30. | (Al-Fahoum, Al-Hmoud & Al-Fraihat, 2013) | IO | S | IR | F, L-R | VWC | 0.1–1.5 m | × | × | HH, HM | MC | ✓ | ✓ | E− | S− | × | PB/S | AV |
| 31. | (Mehta, Alim & Kumar, 2017) | IO | S | US | F, D | VWC | 0.01–2.2 m | 0.01–2.2 m | 0, 40°/15° | HH | MC | ✓ | × | E− | S− | × | PB/S | AV |
| 32. | (Khampachua et al., 2016) | IO | M | US, Acc | F,D | VWC | 0.2–5 m | × | × | HH | MC, SP | ✓ | × | E− | S− | × | × | AV |
| 33. | (O'Brien et al., 2014) | IO | S | US | F, U | WCH | 0.2–1 m | 0.2–1 m | × | CM | MC | ✓ | × | E+ | S− | × | × | AV |
| 34. | MY 2nd EYE (Kassim et al., 2011) | IO | S | IR | F,D, L-R | WCC | 0–1.5 m | × | ×/30° | CM | MC | ✓ | × | E− | S− | × | × | V |

Khan et al. (2018), *PeerJ*, DOI 10.7717/peerj.6058

Peer J

Khan et al. (2018), *PeerJ*, DOI 10.7717/peerj.6058

**Table 2** (*continued*)

| S. No. | Study/Name of the system | OE | Sensors | | | PSU | Sensor Sensing Range | | | OM | CD | System Functionality | | | | LT | User-System Interaction | |
|---|---|---|---|---|---|---|---|---|---|---|---|---|---|---|---|---|---|---|
| | | | S/M | T | SF | | FR | UR | ACA/AP | | | OD | R | E | S | | I | O |
| 35. | (*Rizvi, Asif & Ashfaq, 2017*), | IO | M | US, GPS | F | VWC | 0.02–4 m | 0.02–4 m | 15°/× | HH | MC | ✓ | × | E– | S– | GPS | × | AV |
| 36. | (*Saaid, Mohammad & Ali, 2016*) | IO | S | US | F,D | WCH | 0–2 m | 0–2 m | 90°/× | CM | MC | ✓ | × | E– | S– | × | PB/S | A |
| 37. | (*Daudpota et al., 2017*) | IO | M | US, GPS | F | WCH | 0–4 m | 0–4 m | × | CM | MC | ✓ | × | E– | S– | GPS | × | V |
| 38. | (*Bernieri, Faramondi & Pascucci, 2015*) | IO | M | US, Acc | F | WCH | 0.2–4.5 m | 0.2–4.5 m | 15°/× | HH | MC | ✓ | × | E– | S– | × | × | V |

**Notes.**

OE, Operating Environment; Sensors, (S/M, Single/Multiple; T, Sensor Type; SF, Sensor Facing); PSU, Positioning of the Sensory Unit; Sensor Sensing Range, (FR, Front Range; UR, Upward Range (trunk/head level); ACA/AP, Angular Coverage Area/Angular Placement); OM, Operation Mode; CD, Computational Device; System Functionality, (OD, Obstacle Detection; R, Recognition; E, Echolocation; S, Shorelining); LT, Localization Technology; User-System Interaction, (I, Input; O, Output).

[a] http://smartcane.saksham.org/overview/.
[b] http://www.gdp-research.com.au/index.html.
[c] https://www.ultracane.com/about_the_ultracane.
[d] http://www.palmsonar.com/jp/index.html.
[e] Ray—the handy mobility aid!: http://www.caretec.at/Mobility.148.0.html?&cHash=a82f48fd87&detail=3131.
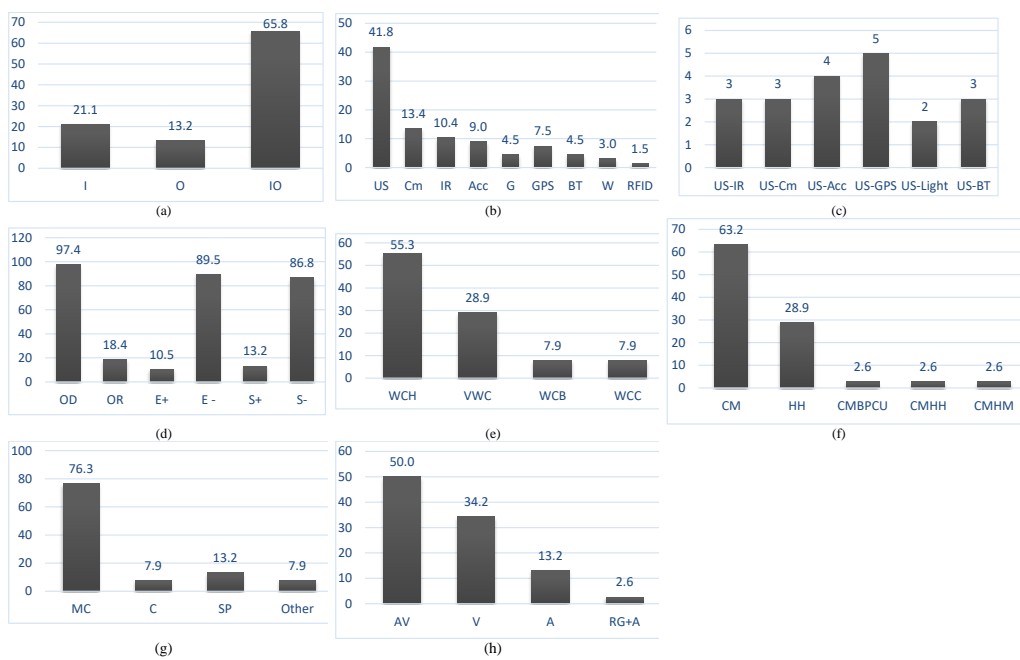

**Figure 3** **Trends in literature for technology-assisted shite cane.** (A) Operation environment (B) individual sensor occurrences; (C) sensors used in combination; (D) system functionality; (E) positioning of sensory unit; (F) device mode of operation; (G) computational devices; (H) user-system interaction.

sensor (US) to detect objects coming in front of the users. *Pyun et al. (2013)* use three ultrasonic and one infrared sensor to detect the extent of an obstacle. Figures 3B and 3C divide the use of sensors into sensor type and single vs. multiple, respectively. Most of the studies used multiple sensors in different combinations including, e.g., US with IR, US with GPS, US with Cm, US with IR and Water Sensor. The most frequently standalone sensors include Cm and US. Sometimes a single type of sensor is used in different combinations, e.g., to sense in multiple directions. However, the trend is towards using single or multiple types of sensors in detecting obstacles in front, left, right, and head level of the blind user. It means that the system should assist the blind users in detecting obstacles not only at the front but also at the trunk level. In addition, each sensor is exploited for a different purpose: US detects obstacles in front of the user, infrared for inclinations, stairs and distance estimation, camera for obstacle recognition, laser for range estimation, a color sensor for color sensing which is used to differentiate the navigation path from the surrounding environment , tilt angle for detecting the slope angle and RFID tags for identifying essential landmarks. Among these sensor, US is widely used (29 out of 38 systems), followed by camera and then infrared, while the remaining ones are used only in specific situations, e.g., GPS for localization, water sensor for detecting water on the ground or puddles, gyroscope for orientation, accelerometer for step detection, etc. The studies encourage the use of multiple types of sensors in combination, which may potentially increase the perception of blind users regarding their surroundings. However, the relationship of the selection of sensors with the three primary techniques associated

with a white cane is not that simple. For example, a user can quickly benefit from their contact with the ground (shorelining) and the interpretation and classification of sounds (echolocation) through the white cane tapping. The situation becomes different when it comes to obstacle detection, where not only the identification of the type of object is an issue but also their position and distance from the blind user. Therefore, the selection of the type and number of sensors should be made in a manner that has no effects on the first two techniques but improves the ability of obstacle detection and extends the range beyond the physical length of the cane. Only a few studies (*Bouhamed, Kallel & Masmoudi, 2013*; *Nada et al., 2015*; *Okayasu, 2010*; *Pyun et al., 2013*) attempted to supplement the shorelining technique only. The design of the virtual white cane needs attention, as the nonavailability of the physical cane affects the echolocation and shorelining capabilities of the users, although they do support obstacle detection due to their inherent design for detecting obstacles.

## Positioning sensors on white cane

Sensors can be mounted at different positions on the white cane. The positioning of the sensors affects the performance of the whole system as well as the use of a white cane. *Nada et al. (2015)* place one ultrasonic sensor near the bottom of the cane for detecting obstacles in front of the user and another near the handle to detect obstacles at head level facing upward. A water sensor is also placed at the bottom of the cane to detect water and puddles on the ground. *Mutiara, Hapsari & Rijalul (2016)* place one sensor at cane handle for detecting trunk-level obstacles and the other at the cane bottom with proper angular placement for detecting holes and hitches.

Virtual white cane is a blind assistive aid designed with the possible characteristics of traditional white cane by using a range of sensors, to collect surrounding data, and a computational unit for data analysis, processing, and presentation to the blind user (*Vera, Zenteno & Salas, 2014*). *Croce et al. (2014)* use a smartphone camera for navigation by reading and analyzing the color patterns drawn on the floor using taped colored lines. The detected color lines are drawn on the screen and identified by the user while sweeping their finger on the screen. The smartphone generates vibrations, when the finger touches the line on the screen, telling the user that they are going in the right direction.

Table 2, column 5 (Fig. 3E) contributes the evaluation of the chosen systems based on the criterion of where the sensor unit is on the white cane. The trend is either supplementing the white cane by mounting multiple sensors or adapting handheld devices as a virtual white cane. For obtaining the maximum sensing capability, three familiar places on white cane have been considered, which include the cane handle, center, or bottom. Most of the studies use the cane handle (upper part) (21 of 38, 55.3%). However, for a more reliable solution, the weight, shape, security, assembly, and folding of the white cane should be considered while placing the sensor unit on the white cane. If possible, the sensing unit should act as a plug-and-play unit, which can be easily attached to the white cane when required and removed otherwise (*Hersh, 2018*).

## Sensors sensing range and coverage

The type of navigational environment impacts the selection of sensors, their sensing capability (precision), power consumption, portability, usage, range and coverage with specified angular placement (*Ercoli, Marchionni & Scalise, 2013*). A blind user finds it difficult to detect obstacles that lie beyond the range of white cane. The sensing range of ETAs should not exceed a certain level to avoid detecting irrelevant or unnecessary objects (*Mehta, Alim & Kumar, 2017*). The cane-mounted or hand-held sensors sense the front (head- and ground-levels) and measure the covered area in degrees. *Saaid, Mohammad & Ali (2016)* carried out experiments to identify a suitable position for the angular placement of sensory unit on a white cane. They placed sensors on the upper and bottom portions of the cane respectively (90° , 45° ) and found that the upper position of the white cane with 90° is the best place for positioning sensors.

Table 2, column 6 summarizes the sensing range (in meters (m)) of the sensors which can be divided into three sub-criteria including front range (FR), upwards range (UR), and angular coverage area with angular placement (ACA/AP). The sensing range is 0.1 m to 5 m for FR and 0.2 m to 4 m for UR, however, few studies mentioned the value for ACA only and others for AP only (see column 6 in Table 2), e.g., FR and UR sensors are used based on user needs, sensor capabilities, and the nature of the target environment. The intended use of these sensors is to warn the user about the obstacles incoming in their way, and therefore, the sensing range can be made limited to a specified range to avoid detecting foreign objects. Also, some of the systems in Table 2 come up with mode changing capability, e.g., Miniguide, which enables the user to switch mode for working in the indoor/outdoor environment by adjusting its sensing range.

## Operation mode

The operation mode (Table 2, column 7) specifies whether the assistive tool is a standalone, cane mounted, or backpack processing unit. Considering the specific needs of blind users, the targeted operating environment, and the functionalities, researchers exploited several operation modes. These include standalone handheld, cane-mounted, cane-mounted with a hand-held device or cane-mounted with back-pack computation unit. Table 2 (Fig. 3F) shows that the majority of the existing solutions (24 out of 38, 63.2%) use a white cane with the mounted sensory unit without requiring any additional device. Only a few considered handheld device (about 28.9%) or and keeping the computational unit as backpack device on the back of the blind user.

## Computational devices

The computational devices (Table 2, column 8) process the sensed data to make it usable for navigation. These devices can be smartphones, computer, microcontrollers and analog circuits. The selection of a specific type of computational device depends upon speed, memory, Analog-to-Digital (AD) conversion, and power consumption. Figure 3G shows that the majority of cases used a microcontroller (29 out of 38, 76.3%), followed by the computer as a backpack (3 out of 38, 7.9%) and smartphone (5 out of 38, 13.2%). Multiple computational devices were also used, e.g., in *AlAbri et al. (2014)*.

## Functionality

An ETA has one of the four possible uses including obstacle detection, its avoidance, navigation, and localization. Researchers have exploited these functionalities in several combinations, e.g., obstacle avoidance with localization. The desired functionality is offered by exploiting technologies in a multitude of ways including, e.g., tags for localization, smartphones for obstacle detection and navigation. Several solutions offer additional functionalities, e.g., *Al-Fahoum, Al-Hmoud & Al-Fraihat (2013)* recognize the material of obstacles such as wood, plastic, steel, and glass. *Ye, Hong & Qian (2014)* recognize the obstacles along with estimating cane pose using Visual Range Odometry (VRO) method. *Batterman et al. (2017)* connected a tactile button to the handle of the white cane for accessing iOS navigation apps such as Blind Square and Seeing Eye, where the input chosen by the blind users is spoken loudly by the VoiceOver app to assist them in making correct selections. The benefit of adopting this approach is accessing the smartphone without holding it. *Rizvi, Asif & Ashfaq (2017)* designed a wearable glove to detect obstacles at user front and by using GPS communicates the location of the blind user in the form of text messages.

Table 2, column 9 (Fig. 3D) shows that the majority of these systems (23 out of 38, 60.5%) detect obstacles at the user's front, left, right, or head level but are unable to offer additional details (hidden semantics) regarding the surrounding objects. The trend is toward mitigating the issue of obstacle avoidance. From column 9, it can be understood that obstacle detection is among the core functionalities and basic techniques associated with a white cane, which have been given significant attention as well as priority in the design of ETAs. In almost 97% of the studies, the leading research design is around the obstacle detection. About 18% support obstacle recognition, whereas echolocation and shorelining attracted few studies. This is in line with our discussion on the selection of sensors in 'The Use of Sensors' that users are comfortable with echolocation and shorelining and that obstacle detection should be focused. Among the studies that implicitly consider echolocation and shorelining are presented in Table 2, where E+ stands when a given study affects echolocation (enhances or degrades it) and E− otherwise. On the other side, S+ means that the designed ETA has a positive impact on shorelining and S− otherwise. However, targeting obstacle avoidance alone will not resolve the issue of assisting blind users in navigation as more contextual information including, e.g., the size of the object, its shape, and type is required to build a mental model of the surroundings. Only a few studies focused on obstacle recognition, localization, echolocation and shorelining.

## Localization technology

Localization is used to determine the current user position and direction (*Hersh & Johnson, 2008*). Several localization technologies are used. These include QR-Codes, GPS, and RFID. Table 3 summarizes technologies used for the localization of blind and visually impaired people and reports on their operating environment (indoor, outdoor, or both), accuracy, sensing range in meters (m), data transmission rate in megabytes per second (Mbps), carrier frequency (in hertz (Hz), kilohertz (KHz), megahertz (MHz), and gigahertz

Khan et al. (2018), *PeerJ*, DOI 10.7717/peerj.6058

**Table 3  Technologies used in localization of blind and visually impaired people**

| Technology | Operating environment | Accuracy | Range | Data rate | Frequency | Required infrastructure | Strengths | Limitations |
|---|---|---|---|---|---|---|---|---|
| RFID | Indoor | 0.11 m | 15 m | 0.000008–0.032 mbps | 125 kHz–915 MHz | RFID tags | Reliable readings. Economic, power-sufficient, can send multiple types of data | Complex and expensive active tags. Less memory, short range, and slow reading of passive tags |
| Bluetooth | Indoor | 2 m | 22 m | 1 mbps | 2.4–2.48 GHz | Specific transmitters | Can transfer voice data | Slower reading rate, short distance, slow data flow |
| WLAN | Indoor | 1 m | 30–50 m | 54 mbps | 2.4–5 GHz | WLAN routers | Fast data access | Fluctuating positional accuracy due to reflected signals, privacy and security issues |
| UWB (Ultra-Wide Band) | Indoor | 0.1 m | 15 m | Nil | 3.1–10.6 GHz | Specific transmitters | Low power than conventional RFID tags, easy to filter due to short duration, can easily penetrate through walls and other objects | More complex transmission than RFID regarding the stipulated time interval recognition and implementation |
| NFC (Near Field Communication) | Indoor | 0.1 m | 0.2 m | 0.424 mpbs | 13–56 MHz | NFC tags | Low power-consumption | Short range |
| Infrared | Indoor | 0.07 m | 5 m | Nil | 0.1 Hz | Specific transmitters | Available on a broad range of devices | Affected by light interference, Line-of-sight positioning |
| Vision | Indoor, Outdoor | 0.1 m | Nil | Nil | 3.5 Hz | Nil | Can detect obstacles, objects and people | Computational overhead, privacy issues, less reliable in changing environments, affected by several interferences such as weather, light, etc. |
| Magnetic | Indoor | 1–2 m | Room level | Nil | 120 measurement/s | Nil | Uses the existing building materials for positioning | Short range |

Peer」

**Table 3** (*continued*)

| Technology | Operating environment | Accuracy | Range | Data rate | Frequency | Required infrastructure | Strengths | Limitations |
|---|---|---|---|---|---|---|---|---|
| Audition | Indoor | 0.4 | 5 m | Nil | 100 Hz–20 Khz | Nil | Economic | Low penetration ability, easily gets affected by other sound interference |
| | Outdoor | ≤0.4 m | 3 m | Nil | 100 Hz–20 Khz | Nil | Economic | The external sounds (traffic, footsteps, etc.) affect the echoes coming from tapping the cane, i.e., affect echolocation. |
| Ultrasonic | Indoor, Outdoor | 0.02 m | 10–50 m | Nil | 1–75 Hz | Nil | Detects near and far obstacle/objects accurately | Requires line of sight, signal reflection causes problem |
| GPS/AGPS/DGPS | Outdoor | 1–5 m | Global | Nil | 20 Hz | Nil | Long range/coverage | Less accurate especially in urban areas with tall buildings |

(GHz)), infrastructure required, strengths and limitations. Table 2, column 10 shows that only five studies used GPS, one used RFID tags, and two used QR-Codes.

## User-system interaction

A blind user interacts with an assistive solution in several ways. Regarding this interaction, the input can be push-button/switches, keypad, and joysticks. The sensed data is processed and presented in several output formats including auditory, vibration, or their combination. Table 2, column 11 (Fig. 3H) shows that most of the studies do not specify their data input methods (16 out of 38, 41.4%) in which 12 used push-buttons/switches, three used keypad, and only one joystick). The majority of the systems use auditory with vibration as the output (19 of 38, 50%), followed by vibration (13 of 38, 34.2%), and then auditory (5 of 38, 13.2%). The audio output is complete but gets affected by external sources and questions the privacy of the user (O'Brien et al., 2014). This can be handled if earbuds or cochlear implants are used, but they should not block the echolocation and other essential sounds from the surroundings that aid in understanding the context and help in localization and safe navigation. Vibration needs direct physical contact with the human body for better results and is not affected by external sources including noise (as in audio output) as well as the privacy concerns (O'Brien et al., 2014).

An ETA that replaces the white cane (and has no other means of tapping the surface) affects the capabilities of blind users associated with the use of echolocation significantly. Examples of such devices include virtual white canes, which are hand-held devices with no physical contact with the ground to collect information. In addition, ETAs that produce continuous auditory output (or vibration patterns) may negatively affect the echolocation-based capabilities i.e., obstacle detection and localization of blind users associated with the use of echolocation technique. This is due to the fact that, either the user will be unable to hear other echoes or their attention span gets affected, respectively. Therefore, it is essential to consider the importance of echolocation while designing ETAs so that the blind user may exploit the resulting solution to the fullest. In a similar way, no direct physical contact with the ground, walkways, pavements, etc., makes it difficult for a virtual white cane to detect such surface textures and communicate to the blind user in an informed way. Therefore, researchers need to focus on this issue so that a more navigation-supportive solution could be designed. In this regard, the use of image processing in detecting and capturing different surface textures and analyzing them could be an excellent research opportunity, if considered.

## Other factors

In addition to the nine criteria mentioned in the previous subsections, other factors and aspects need discussion. These aspects include cost, power consumption, weight, size, shape, ergonomics and the social acceptance of the designed ETAs. These aspects are briefly discussed in this Section.

The concept of "cost" is many-faceted in that it can be defined by several things including the price of sensors, their accuracy, power consumption, the number of sensors to be employed, processing unit/computational devices, and the possible effect of these

on the weight, shape, and size of the white cane. Table 3 points out several important details of these sensors, which together with these factors may potentially define the cost of the reported systems. For example, in Table 2, the cost of each of these systems can be established by looking at column 3, 4, and 8, i.e., the operating environment, use of sensors and the computational devices, which have a direct impact on the price, shape, size, weight, and a power consumption of the resulting white cane solution.

As already discussed, a solution that is designed for both indoor and outdoor environments will be cost-effective as users do not have to use/buy multiple devices to deal with each environment separately. Regarding the effect of sensors on the cost of the white cane, solutions that use less number of low-priced but efficient sensors (real-time processing with low overhead on the battery) will be considered as cost-effective. For example, the US, IR, and inertial sensors provide average accuracy with less power-consumption (3–5 v), and small size can be efficiently and cost-effectively used with a white cane. In contrast, RFID tags and readers are expensive in the case when they are used in large number to cover more area (technically not feasible especially for outdoor environments). Although GPS and Camera sensors give pinpoint accuracy (especially Camera), they are more power-hungry, involve comparatively greater processing overhead, and are expensive than the others. Therefore, the selection of type and number of sensors should be made carefully so that the targeted system operates equally efficiently in indoor/outdoor environments, multiple weather conditions, with low overhead of processing, power, price, and ergonomics. Above these, in the case of commercially available solutions, their price also makes a difference regarding cost. For example, SmartCane (52$) is much cheaper than UltraCane ($1,295) and MiniGuide (545$) and offer almost the same functionalities.[1] Finally, a computational device that is light, smart in size, shape, and weight, and energy-saving is preferred to reduce the cost of the targeted solution further. For instance, SmartCane and Miniguide use a microcontroller as a computation device and therefore, cost-effective regarding low-processing overhead, a minimal number of US sensors, and low power consumption.

The battery life is an significant criterion for ETAs, especially in outdoor environments when the user is out for a long time. However, it can even be problematic in indoor environments in regions where power-outage occurs frequently. Therefore, solutions that are power-hungry become a natural rejection for the users in such circumstances. A solution can be power-hungry if it uses comparatively more sensors than the others and it involves greater processing overhead regarding the number of sensors, type of sensors, computational devices, functionality, user-system interaction methods, etc. In Table 2, solutions that use GPS and Camera are more power-hungry than US and IR. Also, solutions that use multiple sensors than a single or two would be more power-consuming, see, e.g., *Niitsu et al. (2012)*; *Alshbatat & Ilah (2013)*. Moreover, solutions that use a laptop as the computational device will hardly run for 3–4 h, more tiring than a microcontroller, which runs for many hours. For instance, Kinetic Cane and a co-robotic cane are more power hungry because of the number and type of sensors as well as the computational unit than UltraCane, which is less power-hungry.

The weight, size, and shape are other essential yet related factors that need consideration especially when it comes to the use of a white cane. These aspects affect the natural use of a

[1] The authors collected these prices on May 23, 2018, and they are subject to change.
white cane and can be ergonomically and cognitively stressful for a blind user if not handled with care. For instance, using a computational device directly on the white cane or the daily usage device which does not label the user as visually impaired will be much efficient and blind-friendly in contrast to the back-pack computational devices or other bulky hardware even if it has no effects on the weight, size, and shape of the white cane. Also, these factors have a direct effect on the social life of a blind user, e.g., wearing bulky hardware with a white cane may give strange looks to the blind user with low social acceptance especially for children playing in the surroundings. Therefore, a solution should be comfortable for the blind user in terms of size, shape, weight, as well as its social acceptance in the society so that the blind user feel confident while navigating in the surroundings.

In Table 2, Kinect Cane is larger and heavier than Miniguide and Palm Sonar as the former uses the laptop as computational backpack device than the latter ones that use a microcontroller in the form of a handheld or pluggable unit on the white cane. Regarding the shape, Kinect Cane looks more awkward and difficult to manage than Miniguide as the former uses the camera, Microsoft Kinect sensor and accelerometer sensors and laptop computer as computation device than the latter, which uses ultrasonic sensors and microcontroller for computation. It is, therefore, necessary that the size, shape, and weight of the resulting ETA should not affect the utility and salient features of the white cane, which include folding, water resistance, ease in maintenance and conversion (back) to the white cane (when desired). Also, the ETA should be shockproof. These features could be better achieved in ETA by exploiting the handle of the white cane as compared to other areas in traditional white cane.

The last but not the least factor is the issues of *ergonomics* and *social acceptance* of the ETAs. The introduction of bulky hardware to be positioned on the white cane or human, in either case the user will quickly get tired and refuse to use it. The ETA should minimize the learning curve and help the user to quickly adopt it for several activities with minimal information and cognitive overload. The user-system interaction should be designed carefully so that it may not affect the walking speed of the user; if possible, the hands should be free (for in case holding/carrying items such as food), with full attention for the recognition and avoidance of dangerous obstacles (*Hersh, 2018*). The output can be vibration, haptic or auditory. However, the continuous auditory output may potentially overburden the user with the chances of being interfered with the elimination of natural cues from the surrounding such as traffic noise and echoes from the white cane. This can be handled if earbuds or cochlear implants are used, but they should not block the echoes and other essential sounds from the surroundings that aid in understanding the context and help in localization and safe navigation. Also, the ETA should not affect echolocation, shore-lining and other vital clues that are possible when the white cane is used alone. The vibration and haptic feedback could be a better alternative provided that the optimal number of patterns is used so that the user is not overburdened with information and cognitive overload. According to *Lopes et al. (2012)*, the ETAs should ensure the safety of the user during navigation especially crossing the road, passing through heavy traffic, and making visible the user to others using a blinking LED light (when the sensors sense the dark). In addition, for extended outdoor trips, they proposed that the device could be

charged using photovoltaic solar cells. However, these aspects should be considered with care so that these can be socially accepted and may not be an unnecessary hazard for the blind and visually impaired people. Overall, the ETAs should be able "to create a rough mental picture of the surrounding environment, where relevant information is efficiently transmitted without affecting the ergonomics, safety, and social acceptance of blind and visually impaired users, so that they can navigate freely, independently, and confidently in the society.

## CONCLUSIONS

The white cane is among the most widely used navigation aids for blind people. The advancements in ubiquitous modern technology have introduced numerous opportunities for enhancing the utility of white cane. In this regard, researchers have put forward their endless efforts to make technology-assisted white cane useful, reliable, and user-friendly. The literature review presented in this paper is an attempt to discuss what researchers have achieved in the form of technology-assisted white cane (primarily during 2010–17) and what is yet to be achieved. Some of the research trends identified in this literature survey include:

- The majority of the systems presented in this survey use a white cane as the primary assistive tool, being supplemented with several state-of-the-art technologies.
- The majority of researchers attempt to develop a generalized technology-assisted white cane that is capable of adapting itself to the changing navigation environment of the blind user and offer equal benefits both for indoor and outdoor.
- The current research focuses significantly on obstacle avoidance and pays little attention to the hidden semantics of the obstacle such as the size, shape, nature, etc.
- The current research uses the ultrasonic sensor in most cases (43.8%), which show its potential in developing more reliable and user-friendly ETAs especially technology-assisted white cane. The handle of white cane is considered the best point for placing sensory, computational and I/O unit. Also, the virtual white cane is an emerging trend. The preferred input method is pushbutton/switches and output is auditory + vibration.
- Using a separate data processing and the computational unit is considered overhead, and therefore, research is shifting towards using a generalized device capable of both sensing as well as processing the sensory data. In this regard, smartphones can be exploited to a greater extent.
- Most of the reported studies use a microcontroller for computation and decision-making because these were able to readily accept and process the sensory data of the sensors used in combination with it. The generated output is in the form of vibration, beeps or buzzers, which leads to low-battery consumption because of the low processing overhead. In case the image processing is required with high accuracy, backpack units such as laptops are used. However, the trend is shifting towards using smartphone devices due to wider availability and acceptability.

The state-of-the-art literature highlights several future avenues of research. These include the following.
- Researchers should make the end users as part of the design process to achieve a more efficient, customizable, portable, indoor/outdoor adaptable and user-friendly technology-assisted white cane (*Gallagher et al., 2014*; *Hersh, 2018*). This active participation will help us identify the strengths and limitations of existing technologies to develop more robust and reliable navigation solutions, and their input can be better captured and exploited in developing more adaptive and universal ETA solutions especially technology-assisted white cane.
- The cane orientation and sweeping frequency are two essential parameters that define the utility of long cane and therefore, should be considered in designing ETAs (*Kim et al., 2016*). The technological travel aid must be silent and unobtrusive except when generating output (*Pyun et al., 2013*).
- The output produced by the travel aid should be as much tactile as possible. In the case of auditory, it should neither interfere with other audio signals nor overload the auditory sense of the user (*O'Brien et al., 2014*; *Pyun et al., 2013*). In the case of vibration, the device should be designed in such a way that it maximizes the direct physical contact with the end user for better communication.
- The technological travel aid should be capable of error tolerance (*Yelamarthi et al., 2010*).
- A paradigm shift can be observed in blind assistance from simple obstacle avoidance devices to more advanced navigation and wayfinding solutions with the help of smartphones (*Hersh, 2018*). Today, smartphone apps are produced for assisting blind users to offer more contextual details to the blind users, which include, e.g., identifying bus stop, best route, and road crossing with greater control of the blind users on the information produced.

In addition to the above, the following lines present research avenues/recommendations that are not mentioned directly in the analyzed 36 papers (38 systems in total) but we have distilled by carefully, critically, and analytically reviewing them.

- An ETA detects obstacles, provides orientation, localization and signage information to the user for safe and efficient exploration of the environment. However, it is unable to fully adopt, mimic and extend the benefits of the white cane. Therefore, its design needs to be user-centric, portable, reliable, trust-worthy, lightweight, and low-cost, less power hungry with the minimum training. Also, it should exploit the state-of-the-art technologies to either supplement or replace the white cane and enable the user to explore the surroundings quickly. In extending the traditional white cane with technologies, the solutions proposed should not affect its use, weight, shape, security, assembly, and folding. The learning curve should be kept as minimum as possible.
- A technology-assisted white cane should be assessed from the perspective of traditional white cane to evaluate its effectiveness in replicating the available functionalities as well as providing additional ones.
- Regarding the three basic techniques of the white cane, introduced in the Introduction, the obstacle detection has been targeted in most of the available solutions with little attention to echolocation and shorelining. One possible reason could be the inherent

importance of the obstacle detection and recognition, which can act as the first step for the latter two techniques. However, for a complete and effective solution, the three aspects should be considered in the design of ETA.

- If the purpose of an ETA is to enhance (and not replace) the capabilities of the white cane, then the design should allow hearing and responding to the echoes coming from tapping and sweeping the surface by blind users for echolocation and shorelining. This way the blind users may get the maximum benefits from these techniques, which are currently associated with the traditional use of white cane only.
- The ETA design should assist blind users in navigation and extend their sensing range to explore more by providing a complete mental model (contextual details) of their surrounding environment.
- The sensing range of sensors should not exceed to detect irrelevant (and not dangerous) obstacles; in addition, sensors should be appropriately placed on a white cane so that its sensing range is kept unaffected.
- The sensory and computation unit should be cane-mounted without involving any external handheld or backpack hardware units.
- A technology-assisted white cane should come with adaptive user interfaces with simple, robust and real-time input/output generation.
- Smartphones are the ubiquitous and general-purpose portable devices that have been exploited by several researchers in extending traditional navigational aids. Therefore, it is recommended to exploit its capabilities to the fullest in making technology-assisted white cane smarter and reliable.
- Besides the nine criteria, the design of the ETAs should be aligned with other factors discussed in detail in 'Other Factors'. These include the cost, power consumption, size, weight, shape, safety, ergonomics, and social acceptance of the blind users when they use the design solution for localization, obstacle detection and avoidance, and navigation.

The design of ETA should adhere to the following six guidelines issued by the US National Research (*Council, 1986*) that identify what information should be available from an ETA to the blind users. These include:

- Information regarding the presence, location, and nature of the obstacles lying ahead of the bind user.
- Information regarding surface or path, which includes gradient, texture, upcoming steps and left as well as right boundaries.
- Information describing the objects that lie to the left and right sides of the blind user, which includes doorways, fences, hedges, the shoreline of the part.
- Information regarding the presence of aiming point in the distance, which includes traffic sounds and knowledge of absolute or relative travel direction.
- Information regarding location and identification of landmarks such as building entrances, room numbers, elevators, etc.
- Information that is sufficient enough to build a mental model or map of the surroundings of the individual so that they can move confidently.

Therefore, ETAs should be designed while keeping in mind these aspects so that a more efficient, blind assistive navigation aid could be produced. Limitations of this review article include coverage of research articles related to the issues of power-consumption, stipulated response time, and commercial availability of the technology-assisted white cane in more detail. Also, it covers no research papers published in languages other than English. In conclusion, the available technologies have the potential to be fully exploited in making white cane more useful. Researchers have already achieved some milestones; however, exploiting their potential to the fullest in designing a more reliable, portable, user-friendly, robust, and more assistive white cane is yet to be achieved. This research paper attempted to identify such future possible endeavors and invites researchers and practitioners to make the lives of blind users happier and healthier.

### Funding
The authors received no funding for this work.

### Competing Interests
The authors declare there are no competing interests.

### Author Contributions
- Izaz Khan, Shah Khusro and Irfan Ullah conceived and designed the experiments, performed the experiments, analyzed the data, contributed reagents/materials/analysis tools, prepared figures and/or tables, authored or reviewed drafts of the paper, approved the final draft.

### Data Availability
    The research in this article did not generate any data or code; this is a literature review.

### Supplemental Information
Supplemental information for this article can be found online at http://dx.doi.org/10.7717/peerj.6058#supplemental-information.

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
