# Peer review of "Technology-assisted white cane: evaluation and future directions"

_PeerJ, doi:10.7717/peerj.6058_

## Round 0.1 · original submission · Major Revisions

· Academic Editor

Major Revisions

Two reviewers read the manuscript. One felt that the paper wasn’t novel and needed significant improvements to the English and the other thought it was worthwhile and met PeerJ requirements, though they suggested several major revisions. I agree with the latter reviewer. Please follow all advice from both reviewers, and have the manuscript carefully edited by somebody with professional-level English writing skills, before resubmitting.

Reviewer 1 ·

Basic reporting

This is a review of literature for rather small time frame of 2010 to the present. It is not clear why the authors chose such a limited amount of time since there is a lot of literature dating back to the 1970s which is just as relevant today as what has been cited. As one example the laser cane which was actually mass produced and introduced to the market in the 1970's is relevant to this topic. Another is the Sonic Guide also produced in the 1970s. Of course there are other technologies from the 1980s and 90s going forward. The Pathfinder as one example. The reason these products are relevant is because they covered the same approach to technology as what has been completed in the past 10 years. While certainly sensors have become more advanced in information acquisition, processing speed, they are smaller and less expensive, the underlying fundamentals of what these technologies do and don't do has not changed for the past 40 years. So if the authors want to argue that there are changes to the underlying fundamentals the case has not been made.

Experimental design

This is a literature review.

Validity of the findings

I question the limited time frame of the literature review. Perhaps there is a strong rationale. It should be stated.

·

Basic reporting

This is a generally well-written review MS aimed at obtaining an inventory of electronic enhancements of the white cane reported over the period 2010 – 2017, and at suggesting which of these (and further) enhancements will maximize the utility of the enhanced cane as a travel, orientation, and obstacle detection and recognition aid for blind users. This is certainly a worthwhile endeavor in view of the multiple approaches to this problem that have been attempted, the wide range of technology available, and mostly the challenge it poses: The white cane, for all its simplicity, provides the well-trained blind person with a wealth of information about his/her immediate surroundings and the obstacles that may be present.
In my opinion, the authors have met the basic requirements of a PeerJ MS, even if I have a number of suggestions for changes that may improve organization, clarity, and wording. The references, figures, and tables are clear and provide the raw data on which their text presentation and interpretation are based.

Experimental design

The authors argue, convincingly, that the topic of their study is important and will help provide direction to the continued development of electronic travel and orientation aids. Their search of the existing literature appears to have been thorough, and although they could have provided slightly more detail about the selection process used to winnow 573 papers down to 32, the remaining ones appear to provide a broad spectrum of tools used to supplement the white can, or in some case replace it with a virtual tool.
The authors have done a very nice job of analyzing the different aspects of the "enhanced canes" and presenting the results in tables that are relatively easy to understand, although the layout could be cleaned up a little, and more detail in the table captions would be helpful.

Validity of the findings

To the extent that the validity of the data from a literature review can be ascertained, it appears that the authors have done a very credible job representing all important aspects of the papers they included, and in my opinion this is what makes the MS very helpful. The only thing that is somewhat disappointing is that there is not much effort on the part of the authors to discuss, cost, weight, power, ergonomics, etc., of the different approaches, even if those are not always explicitly mentioned in the original articles, and subject to change, and don't take these aspects into account in their conclusions (other than by saying that many of the papers did not provide this type of information explicitly). I would encourage the authors to devote more attention to these aspects in the Discussion.

Additional comments

While the MS is generally well organized, improvements are possible on a number of points:
1) The Results section start with a paragraph that, except for the last sentence should replace the opening paragraph of the Introduction, which is much less engaging. Move lines 154-166 to replace
2) What is missing from the Introduction (or Discussion), in my opinion, is a better explanation how the white cane provides echo location and "shorelining" (following a wall or pavement edge) capabilities as well as obstacle detection, and that any auditory output from an ETA should not interfere with the auditory aspects of echo location, traffic noise, and other sound beacons commonly used by cane travelers..
3) Lines 354-381 (and possibly also some of the points in lines 333-351) of the Conclusion section should be moved to a General Discussion, and should be separated into points that were brought up by previous authors and are adopted by the current authors, and points that the previous authors did not make but that the current authors have distilled from integrating these 32 papers.
4) Repeatedly the authors cite privacy concerns as a potential downside of using auditory output, but with sound played into an earbud, hearing aid or cochlear implant this is not really a concern. What is a concern, however, is the potential danger associated with loss of environmental auditory information, so the authors may want to mention safety as an important consideration in favor of tactile output.
5) The presentation in lines 175 – 324 should refer to specific columns in Table 2 (just like it refers to specific panels in Fig. 3), and refrain from putting in too many citations, as these can be easily seen in the table
6) In many places the authors speak of "extending the cane" when they really mean "enhancing the utility of the cane" or "supplementing the functionality of the cane."
7) There are many other instances of awkward wording that should be addressed, I strongly suggest that the authorss have an experienced writer of scientific English comb through the MS after it has been revised.

---

## Round 0.2 · Major Revisions

· Academic Editor

Major Revisions

Two reviewers read your manuscript and reviewer 1 continues to feel that it is poor, but decided to recommend accept instead of reject because I did not follow their advice to reject on the first review. Their only comments (in confidential notes) are that the review remains lackluster due to its minimalist approach to reviewing the literature. Whereas the reviewer feels a literature review should be comprehensive, the authors have limited their review arbitrarily to only very recent studies, which creates significant gaps in the manuscript. I agree that the reviewers should cover the concepts they are discussing all the back to their roots, and limit scope by limiting the number of complete concepts reviewed, rather than by providing a shallow review of many concepts. Reviewer 2 feels the authors have improved the manuscript and has specific advice on further necessary improvements, including clearer discussion of core concepts that are currently conflated. On the next revision, reviewer 2 must be satisfied that the manuscript is ready for publication so please carefully address their concerns completely.

Reviewer 1 ·

Basic reporting

My comments are contained in the section for the editor

Experimental design

My comments are contained in the section for the editor

Validity of the findings

My comments are contained in the section for the editor

Additional comments

My comments are contained in the section for the editor

·

Basic reporting

It is good to see that the authors, at the request of reviewer #1, expanded their search criteria to older publications and added 4 references pertaining to older technological cane solutions that are still being used.

Experimental design

The authors have carefully addressed the concerns of the reviewers and further strengthened the MS.

Validity of the findings

authors have carefully addressed the concerns of the reviewers.

Additional comments

Revision:
The authors have done a commendable job in revising the MS along the lines of both reviewers. I do have a few additional comments, mostly regarding issues that arose out of the text revisions, but these should be relatively simple to address.

Remaining/new comments:
It is possible that the authors are not familiar with some of the terms in my previous review, and the same may be true for readers of the paper. A good example of this is the following sentence: "…detect 47 obstacles in the range of 1.2 meters at ground level using echolocation and shore-lining" This indicates that the authors consider echolocation and shorelining as methods of obstacle detection, whereas in reality these are 3 distinct techniques in cane use. There are 3 basic uses of the white cane:
1) Obstacle detection, i.e., as cane users walk along, they sweep the cane back and forth in front of them, to avoid running into obstacles or tripping over surface ridges, and to detect steps, curbs, and dropoffs.
2) Shorelining, i.e. , following a border, which can be the wall of a building or corridor, the edge of the pavement, or pavement markings created especially to guide blind people in public places, such as bus stops, crosswalks, shopping centers, etc. The user still sweeps, but touches the border on every sweep to keep the distance from the border constant as (s)he goes forward.
3) Echolocation, i.e., tapping the cane during the sweeping motion to hear the echo of a nearby wall, parked car, or any other sound-reflecting surface, and thus gain information about the surroundings, without actually having to touch them.
Adding these 3 concepts to the Introduction will resolve the misconception that seems to be present in Line 48, and also make it clearer in the presentation of different systems and reviews whether each technological innovation does or does not form an extension/enhancement of each of these basic uses. My guess is that very few ETAs will assist with echolocation, but some may be better suited for shorelining than others. Most importantly, in discussing various technologies, the author should ask the question whether the technology could interfere with any of the 3 functions mentioned above, as this would limit rather than enhance the original cane.

Wording comments:
The MS uses many abbreviations and some of them are not introduced in the text (e.g., the term "Cm"); even if they are, this happens at varying points. It would be good to have a list of abbreviations in one location; the caption of Figure 3 would be a good place for this, since they are all used in that figure.

---

## Round 0.3 · Minor Revisions

· Academic Editor

Minor Revisions

One reviewer re-read your article and feels you’ve done a great job towards improving your manuscript. They have a few minor suggestions that you should please make before we accept for publication.

·

Basic reporting

The authors have done a commendable job to address the previous comments by reviewers and editor. It is interesting to see that in a number of instances they have pointed out the importance of retaining the echolocation capabilities of the physical cane in designing ETAs, but have not done the same for the shorelining capabilities. I am pointing out examples below.

Experimental design

No remarks

Validity of the findings

There are a few specific areas where I would like to see a broader interpretation that covers shorelining in addition to echolocation.
1) Line 238-245: I would argue that the shorelining and echolocation techniques don't need ETAs: The physical contact during shorelining and the classification of sounds during echolocation work just fine, so the ETA should not interfere with those, but augment the ability to detect obstacles, and possibly extend the range beyond the physical length of the cane.
2) Line 325-327: Similarly, this statement should be changed to indicate whether any of the ETA devices leave the echolocation and shorelining capabilities intact, and which studies have explicitly included these capabilities in their evaluation.
3) Line 356-364: This is a good argument, but it can be extended to shorelining: It may be difficult to design an ETA that can detect differences in surface texture (such as grass along a walkway or different types of pavement) as well as physical limits, from edges just 1 cm high to walls or fences.
4) Line 539-543: Here again shorelining should be mentioned in addition to echolocation.

The major change I would like to see is in the functionality described and categorized in the tables: Rather than talking about Navigation (N) (which includes shorelining, echolocation and obstacle avoidance, but not much else that I can think of), I would introduce echolocation (E) and Shorelining (S) as specific types of functionality, and evaluate all ETAs explicitly on those, so each cell in the functionality colummn of Table 2 gets an E+ or E- and an S+ or S- to indicate whether this device has those capabilities.
Finally, in Table 3 under technologies Audition (rather than "Audible") should be split into use of external sounds (e.g., listening to traffic patterns and footsteps) and use of sounds created by the device (echolocation); there also should be a technology called Touch, and its role should be listed as obstacle/boundary detection but also texture detection

Additional comments

After everything else has been completed it would be good to get an outside reader with excellent command of English to proofread the MS one more time: There are not many errors, but they are annoying and can easily be found and fixed by an attentive reader.

---

## Round 0.4 · accepted · Accept

· Academic Editor

Accept

Thank you for your submission and for so diligently addressing the reviewer comments.